# *C9orf72*-G_4_C_2_ Intermediate Repeats and Parkinson’s Disease; A Data-Driven Hypothesis

**DOI:** 10.3390/genes12081210

**Published:** 2021-08-05

**Authors:** Hila Kobo, Orly Goldstein, Mali Gana-Weisz, Anat Bar-Shira, Tanya Gurevich, Avner Thaler, Anat Mirelman, Nir Giladi, Avi Orr-Urtreger

**Affiliations:** 1The Genomic Research Laboratory for Neurodegeneration, Neurological Institute, Tel Aviv Sourasky Medical Center, Tel Aviv 64239, Israel; Hilakobo@tauex.tau.ac.il (H.K.); orlyg@tlvmc.gov.il (O.G.); maligw@tlvmc.gov.il (M.G.-W.); anatbn@tlvmc.gov.il (A.B.-S.); 2Sackler Faculty of Medicine, Sagol School of Neuroscience, Tel Aviv University, Tel Aviv 6997801, Israel; tanyag@tlvmc.gov.il (T.G.); avnert@tlvmc.gov.il (A.T.); anatmi@tlvmc.gov.il (A.M.); nirg@tlvmc.gov.il (N.G.); 3Movement Disorders Unit, Neurological Institute, Tel Aviv Sourasky Medical Center, Tel Aviv 64239, Israel; 4Laboratory for Early Markers of Neurodegeneration, Neurological Institute, Tel Aviv Sourasky Medical Center, Tel Aviv 64239, Israel

**Keywords:** Parkinson’s disease (PD), *C9orf72*, intermediate repeats, hexanucleotide expansions

## Abstract

Pathogenic *C9orf72*-G_4_C_2_ repeat expansions are associated with ALS/FTD, but not with Parkinson’s disease (PD); yet the possible link between intermediate repeat lengths and PD remains inconclusive. We aim to study the potential involvement of these repeats in PD. The number of *C9orf72*-repeats were determined by flanking and repeat-primed PCR assays, and the risk-haplotype was determined by SNP-array. Their association with PD was assessed in a stratified manner: in PD-patients-carriers of mutations in *LRRK2*, *GBA*, or *SMPD1* genes (*n* = 388), and in PD-non-carriers (NC, *n* = 718). Allelic distribution was significantly different only in PD-NC compared to 600 controls when looking both at the allele with higher repeat’s size (*p* = 0.034) and at the combined number of repeats from both alleles (*p* = 0.023). Intermediate repeats (20–60 repeats) were associated with PD in PD-NC patients (*p* = 0.041; OR = 3.684 (CI 1.05–13.0)) but not in PD-carriers (*p* = 0.684). The *C9orf72* risk-haplotype, determined in a subgroup of 588 PDs and 126 controls, was observed in higher frequency in PD-NC (dominant model, OR = 1.71, CI 1.04–2.81, *p* = 0.0356). All 19 alleles within the risk-haplotype were associated with higher *C9orf72* RNA levels according to the GTEx database. Based on our data, we suggest a model in which intermediate repeats are a risk factor for PD in non-carriers, driven not only by the number of repeats but also by the variants’ genotypes within the risk-haplotype. Further studies are needed to elucidate this possible role of *C9orf72* in PD pathogenesis.

## 1. Introduction

Parkinson’s disease (PD) is a common neurodegenerative disorder, affecting about 2% of the elderly population worldwide [1]. Its complex genetic background has been revealed in the past decades, implicating many genes associated with the disease. A wide variety of genetic changes and mechanisms are involved in PD, including rare and common variants, recessive, dominant, and oligogenic inheritance, and epigenetics [2,3,4,5,6,7,8]. However, the full range of genetic changes in PD is still evolving.

G_4_C_2_ Hexanucleotide repeat expansions in *C9orf72* are strongly associated with amyotrophic lateral sclerosis (ALS) and frontotemporal dementia (FTD) [9,10], mostly in European and North American populations [11,12]. Although 30 repeats and over are considered pathogenic for ALS and FTD, most patients that are *C9orf72*- associated-ALS or -FTD, carry an expanded allele with hundreds, or even thousands, of repeats [12,13,14]. Interestingly, parkinsonism was observed in more than 40% of FTD and FTD/ALS patients with pathogenic *C9orf72* expansions [15]. This observation has led researchers to investigate the possible association of *C9orf72* expansions with PD.

While rare cases of PD patients with 30 to 60 repeat expansions or more in *C9orf72* were detected (<0.7%), there was no association with PD [16,17,18,19,20]. Few studies have suggested that intermediate-size repeat lengths in *C9orf72* may be a risk factor for PD; however, these studies suggested different repeat lengths for this association: ≥7 repeats in Han Chinese [21], and ≥20 repeats in Caucasians [17]. In a multi-center meta-analysis of mostly Caucasian and Asian populations, the pathogenicity expansions threshold was determined as >60 repeats, and the intermediate repeat size was set as 17–60 repeats [22], suggesting an effect on PD-risk for the cutoff of 17 repeats, and even as little as 10 repeats as a stand-alone or as a cutoff. In a recent comprehensive review by Bourinaris and Houlden [23], *C9orf72* intermediate repeat lengths were reported in several parkinsonism and movement disorders, including in Dopa-responsive PD, atypical parkinsonisms including PSP and MSA, essential-tremor plus parkinsonism, and spinocerebellar ataxia.

As previous studies have shown the pivotal role of genetically homogeneous populations, such as the Ashkenazi Jews (AJ), in understanding the genetic background of neurodegenerative diseases [24,25,26,27], we hereby determined *C9orf72* repeats’ size in PD patients of Ashkenazi origin and examined their potential association with PD. We also studied the possible association of the shared risk-haplotype, which is observed in carriers of intermediate repeats, with PD-risk.

## 2. Methods

### 2.1. Population

This study included a cohort of consecutively recruited unrelated 1106 PD patients of full Ashkenazi Jewish origin (Table 1). Patients were recruited between the years 2005 and 2015. The diagnostic criteria, recruitment, and genotyping for *LRRK2*, *GBA*, and *SMPD1* mutations have been previously described [24,25,28]. Carrier patients (PD-carriers) were defined with one or more of the following mutations in *LRRK2* (p.G2019S), *SMPD1* (p. L302P), or any of the 10 *GBA* mutations (c.84insG, IVS2 + 1G > A, p.V394L, p.N370S, p.L444P, p.R496H, p.E326K, p.T369M, p.R44C, and 370Rec). Patients who did not carry any of these mutations were determined as non-carriers (PD-NC). The cohort of 600 ethnically matched control individuals used in this study has been previously described [26].

### 2.2. Determining the G_4_C_2_ Hexanucleotide Repeat Length in the C9orf72 Gene

To determine the number of repeats in *C9orf72*, flanking and repeat-primed PCR assays were performed as previously described [9,10], with some modifications [26]. This method detects all repeats expansion but can determine the number of repeats up to 55 repeats. Therefore, in all individuals that carried an allele with 30 repeats or more, an additional method was used to determine accurately the repeat number up to 145 repeats (Asuragen assay kit AmplideX^®^ PCR/CE C9orf72 Kit; Asuragen Genetics; Austin, TX, USA).

### 2.3. Assembly of the Risk-Haplotype within the C9orf72 Locus

To determine the presence of the risk-haplotype in *C9orf72*-locus, we used the genotype data (from Vacic et al. [29], Affymetrixs SNP6.0) of 127 AJ controls (all part of our cohort of 600 controls) and 597 AJ-PD patients (594 are part of our cohort of 1106 PD patients). When determination of the presence of the risk-haplotype was impossible due to missing genotypes, these individuals were excluded (Control = 1, PD = 5). One PD was excluded due to low genotype rate, one was not tested for repeat size, and two PDs who carried >145 repeats were also excluded. In total, 126 controls and 588 PD patients were analyzed.

### 2.4. Statistical Analyses of C9orf72 G_4_C_2_ Hexanucleotide Repeats

All statistical analyses were performed using IBM SPSS statistics software v25 (IBM Corporation, New York, USA). Differences in continuous variables were tested using Mann–Whitney U test or *t*-test (2-tailed). To test the difference in *C9orf72* repeat lengths between patients and controls, both alleles repeat sizes (per individual) were included.

Categorical variables were compared using 2-sided χ^2^-test, or Fisher’s exact test when numbers were less than 5. Odds ratio (OR), with 95% confidence interval (CI), was applied to assess the association of *C9orf72* G_4_C_2_ repeat lengths with PD. This association was examined using the longest repeat size (per individual) as the independent variable. Association of the risk-haplotype with PD was examined using a dominant model. Logistic regression analysis was performed when using repeat units as a quantitative trait (the largest allele or the sum of both alleles).

## 3. Results

### 3.1. Allele Frequencies of C9orf72 G_4_C_2_ Hexanucleotide Repeats in Ashkenazi PD Patients

Allele frequencies of *C9orf72* repeats were determined in our cohort of 1106 Ashkenazi PD patients (Table 1), that was divided into two groups based on their genotypic status, either carriers of PD-associated mutations (see Methods section), or non-carrier patients (PD-NC). We ran a stratified analysis based on the carrier status in *LRRK2*, *GBA*, and *SMPD1*, as a high percentage of our PD cohort carry risk alleles in these 3 genes (35.1%, 388/1106), and these carrier-patients may mask the effect of the hexanucleotide repeat length on PD-risk in non-carrier patients. The most frequent alleles found were 2, 8 and 5 repeat units (66.2%, 12.1%, 8.9% in carrier, and 63.1%, 12.7%, 9.7% in non-carrier patients; Figure 1 and Appendix A). These alleles were also shown as the most common alleles in our previously published data of Ashkenazi controls (66.4%, 11.0%, and 10.2%) [26]. No significant difference in allele distribution was observed between patients with mutations (in *LRRK2*, *GBA*, or *SMPD1* genes) and controls (Mann–Whitney *U* test *p* = 0.756; 4.23 ± 6.13, N = 776 and 4.09 ± 5.36, N = 1200, respectively). However, a significant difference in allele distribution was detected in PD-NC (Mann–Whitney *U* test *p* = 0.034; 4.71 ± 8.40, N = 1436). This was also significant for the total number of repeats (combining the numbers of repeats from both alleles; excluding individuals with expanded alleles of >145 repeats) in PD-NC (Mann–Whitney *U* test *p* = 0.023; 8.63 ± 5.57, N = 714, and 7.94 ± 5.01, N = 599) and was not significant in PD-carriers (Mann–Whitney *U* test *p* = 0.565; 8.10 ± 4.90, N = 387).

### 3.2. The Association of C9orf72 G_4_C_2_ Hexanucleotide Intermediate Repeat Lengths with PD in Ashkenazi Patients

We examined the association of the longest repeat allele in each individual with PD (in PD-carriers and PD-NC). First, we examined whether large expansion lengths (>145 repeats) are present within our cohort of PD patients and controls. We found one PD-carrier patient (carrying the *LRRK2* p.*G2019S* mutation, 1/388 = 0.3%), four PD-NC (4/718 = 0.6%), and one control (1/600 = 0.2%) that carried an expanded allele. No significant association was detected in any of the patients’ groups compared to controls (Fisher’s Exact Test *p* = 1.000 and *p* = 0.384, respectively). Of interest, one of these PD patients was later diagnosed with ALS, and two had dementia. For further analysis, we excluded these five carriers with *C9orf72* G_4_C_2_ repeats expansion (all with > 145 repeats). None of our patients or controls carried alleles between 60 and 145 repeats.

Next, we examined if intermediate *C9orf72* hexanucleotide repeats (20–60 repeats) were associated with PD in our cohort. Among the 1101 PD patients, 3 PD-carriers (0.8%, 3/387) and 13 PD-NC (1.8%, 13/714) carried an allele with intermediate repeat lengths, compared to 3 among the controls (0.5%, 3/599). An association with increased risk for PD was observed in PD-NC patients compared to controls (Fisher’s exact test, *p* = 0.041; OR = 3.684, CI = 1.045–12.990; Table 2, Figure 1), but no association was detected when comparing PD-carrier patients to controls (*p* = 0.684, Table 2).

Measuring the effect of the risk associated with an increasing number of repeats (β, regression analysis for each one repeat unit) shows a significant effect in PD-NC when looking either at the largest allele or at the sum of alleles (β = 0.032, *p* = 0.015, OR = 1.032, CI = 1.006–1.059; β = 0.025, *p* = 0.021, OR = 1.025, CI = 1.004–1.047; respectively). No effect was observed in PD-carriers (β = 0.009, *p* = 0.542, OR = 1.01, CI = 0.979–1.041; β = 0.007, *p* = 0.617, OR = 1.007, CI = 0.981–1.033; respectively). No effect on age of motor symptoms onset was observed in PD-NC (*p* value = 0.978).

### 3.3. The C9orf72 Risk-Haplotype Is Associated with Higher RNA Expression Levels and with PD

We have previously shown that the *C9orf72* expansions in AJ shared a risk-haplotype that expands 107Kb (Goldstein et al. [26], hg19: chr9:27484575-27591569), encompasses the complete *C9orf72* gene, and includes 44 informative single nucleotide variants (SNVs), with significant association with higher number of repeats (over 8 repeats). We further examined here the effects of these 44 SNVs (within the risk haplotype) on the RNA expression levels (eQTL) and splicing (sQTL) of their adjacent genes, as reported by the Genotype-Tissue Expression (GTEx) project [30]. Of them, 11 SNVs had no QTLs, 4 had low effect size (absolute NES < 0.2), and one had no significant effect (*m*-value < 0.9). Nine other variants were also excluded due to high allele frequency in AJs (> 0.5 in gnomAD v3.1 database of non-neuro cases). The other 19 SNVs within this 107Kb risk-haplotype were all associated with higher *C9orf72* RNA expression levels compared to the expression levels of the non-risk alleles (Table 3, Figure 2). Cerebellum and Nucleus-accumbens were the tissues with the highest normalized effect size (NES). Moreover, as GTEx evaluates the effect of each SNV on the neighboring genes within a 2 Mb interval (1 Mb upstream and 1 Mb downstream), it is important to note that all 19 SNVs affected exclusively *C9orf72*-RNA levels and not any other genes within that 2 Mb window. These SNVs also affected splice variants (sQTL) in an exclusive manner, only for *C9orf72*, mainly in cerebellum (Table 3).

We, therefore, tested if carrying the risk-haplotype is associated with PD. Genotyping data from 127 AJ controls and 597 AJ-PD patients were assembled (see Methods), and the presence of the 19-SNVs-risk-haplotype was determined. Overall enrichment of the risk-haplotype was observed in PDs compared to controls: 167 out of 588 PDs carried one or two copies of the risk-haplotype (28.4%) compared to 24 out of 126 controls (19.0%). When stratifying based on mutation carrier status (PD-carriers and PD-NC), a significant association was detected in PD-NC: 28.6% of them carried one or two copies of the risk-haplotype (OR = 1.71, CI = 1.04–2.81, *p* = 0.0356), and tendency was shown in PD-carriers (28.0%, OR = 1.65, CI = 0.97–2.82, *p* = 0.0656).

We also attempted to define the correlation between the existence of risk-haplotype and number of repeats, by looking at all alleles (*n* = 1428 alleles): 100% negative correlation existed between the risk-haplotype and the 2-repeats’ allele (with zero percent risk-haplotype), and 100% positive correlation was observed in carriers of 14–60 repeats (100% carried the risk-haplotype, Appendix A). Thus, we used 14 repeats as the best assessor for carrying the risk-haplotype and re-calculated the association of 14 repeats and higher with PD in our cohorts. Among all alleles in PD-NC, 2.3% had 14–60 repeats, compared to only 1.3% in controls, showing a trend toward significance (Figure 3a, OR = 1.75, CI = 0.96–3.19, *p* = 0.069, uncorrected), with no significance observed in PD-carriers (Figure 3b, OR = 1.46, CI = 0.72–2.97, *p* = 0.296). Further analysis showed a significant association in PD-NC when the cutoff of 13 repeats or 12 repeats was selected (OR = 1.70, CI = 1.07–2.72, *p* = 0.0257 and OR = 1.51, CI = 1.01–2.25, *p* = 0.044, respectively, Figure 3a, uncorrected), while there was no significant association in PD-carriers (OR = 1.46, CI = 0.72–2.97, *p* = 0.296; OR = 1.39, CI = 0.79–2.42, *p* = 0.249 and OR = 1.32, CI = 0.83–2.12, *p* = 0.245, respectively, Figure 3b). No significance was shown at 11 repeats cutoff, both in PD-NC and PD-carriers (Figure 3, OR = 1.22, CI = 0.89–1.69, *p* = 0.214 and OR = 1.20, CI = 0.82–1.74, *p* = 0.345, respectively).

## 4. Discussion

More than 40 diseases, most of which affect the nervous system, were identified with a genetic basis of expansions of simple short DNA sequence (reviewed by Reference [31]). Among these diseases are myotonic dystrophy, Huntington’s disease, spinocerebeller ataxia (SCA), spinal and bulbar muscular atrophy (SBMA), and Fragile-X syndrome. A common finding in these disorders is the correlation of the number of repeats with clinical phenotypes and penetrance. Interestingly, emerging studies suggest that expanded repeats and intermediate repeats can cause or act as risk factors for different neurological diseases, depending on the number of repeats. This was suggested for the *FMR1* gene, when over 200 CCG repeats cause mental retardation (Fragile-X syndrome), while the premutations of 45–200 repeats are a risk factor for Fragile-X- tremor/ataxia syndrome (FXTAS) in males and premature ovarian failure 1 (FXPOI) in females. Other examples are *ATXN2*, in which more than 34 repeats cause SCA2, while 29–33 are risk factor for ALS [32,33], and *ATXN1*, in which more than 38 repeat cause SCA1, while ≥33 are risk factor for ALS, mostly in *C9orf72* expansion carriers [34]. The latter is an example of the complexity of these mechanisms, as some of these risk alleles are more significant in specific subgroups of patients. To see if a similar phenomenon exists also for *C9orf72* intermediate repeats, we analyzed Parkinson’s disease patients of Ashkenazi origin, in a stratified manner, in carriers of mutations in *LRRK2*, *GBA*, and/or *SMPD*1 genes, and in patients that do not carry these mutations. We showed that the expanded alleles (>60 repeats) had no association with PD, as shown by other groups and in a meta-analysis [16,17,18,19,20,21,22]; however, intermediate-size repeats of 20–60 are significantly enriched in PD-NC, increasing the risk for PD, while, in PD-carriers, there was no effect. The high significant odds ratio of 3.68 in non-carriers may be due to the exclusion of those patients who carry known risk alleles in LRRK2, GBA, or SMPD1, as we believe that in these patients the risk for PD is likely influenced by these mutations and not by the C9orf72 intron 1 hexanucleotide repeat numbers. Of note is that Xi et al. also reported that intermediate repeats of 20–29 were found only in PD-NC, and not in PD-patients that carry the *LRRK2* p.*G2019S* mutation [19]. Based on these observations, we propose that *C9orf72* G_4_C_2_ hexanucleotide repeats in intron 1 act as a risk factor for PD when number of repeats are intermediates. Although the exact definition of intermediate- and pathogenic-length size is still in debate, and may differ in different populations, we believe that the presence of alleles lower than the 60 repeats, in the intermediate range, should not be dismissed as a potential risk for PD.

How may intermediate numbers of repeats affect the risk to develop PD?

In ALS, although the role of *C9orf72* expansions is not yet fully established, three main mechanisms are proposed to contribute to its pathogenicity: *C9orf72* loss-of-function, generation of toxic RNA aggregates, and short peptide accumulation [12,14]. As G_4_C_2_ repeats in the *C9orf72* gene are located within intron 1 and are near the promoter region, they may lead to changes in promoter regulation, depending on the number of repeats. Indeed, it was shown that ALS/FTD-expanded alleles are highly methylated and lead to lower levels of *C9orf72* mRNA and protein [35,36,37].

Do these same mechanisms contribute to Parkinson’s disease risk? We demonstrated here that the risk-haplotype, which is shared by the intermediate *C9orf72* G_4_C_2_ repeats, includes many SNVs that have the same effect of increasing RNA expression levels of *C9orf72*, as reported in GTEx. We, therefore, suggest that the mechanism involved in the effect of intermediate-size repeats on PD-increased-risk, could be the higher expression of *C9orf72*. This mechanism was recently suggested for a different neurodegenerative disease, Corticobasal Degeneration (CBD), a rare neurodegenerative disease that shares some similar clinical features with PD [38]. Researchers showed a significant enrichment of intermediate repeats in autopsy-proven CBD, as well as increased *C9orf72*-RNA expression levels in human brain tissues and in CRISPR/cas9 knock-in iPSC cells, but no association with pathologic RNA foci or dipeptides aggregates.

One important question is whether intermediate-repeats-sizes or the increase of *C9orf72* expression, which is associated with the risk-haplotype, may drive the risk for PD. As these two mostly go together, this question should be answered in an experimental set-up that separates the two events. Cali et al. tried to answer this question by knocking-in 28 repeats into iPSCs that normally have either 2 or 6 repeats (suggestive of cells that do not carry the risk haplotype) and demonstrated that these knocked-in cells show higher expression of *C9orf72* [38]. As GTEx data suggest higher expression of *C9orf72* for all 19 variants within the risk-haplotype, it is tempting to suggest that PD-risk may be determined by the level of *C9orf72* expression, mediated by both the risk-haplotype and the number of repeats, a hypothesis that needs further evaluation. This hypothesis raises other questions: does the overall level of *C9orf72* expression depend on the effect of the total number of repeats in both alleles, and whether the genomic region of the *C9orf72*-risk-haplotype might expand to a larger interval than the minimal linkage disequilibrium of 107 Kb, as suggested by gnomAD database? In addition, the GTEx data show that the risk-haplotype effect on *C9orf72* RNA expression levels is not uniform in all tissues. The effect size is large and significant mostly in brain tissues, as well as in the small intestine, but much smaller in whole blood and lymphocytes.

What could be the effect of intermediate-size repeats on cellular expression? Cali et al. performed a comparative genes expression analysis between cells with intermediate repeats and cells with low number of repeats [38], demonstrating upregulation of genes that are enriched for vesicle trafficking and protein degradation pathways, including golgi vesicle transport, response to ER-stress, and autophagy, pathways which are involved in PD.

In summary, our stratified analysis suggests that intermediate- size hexanucleotide repeats in *C9orf72* are a risk factor for PD in individuals who do not carry common AJ founder mutations in *LRRK2*, *GBA*, or *SMPD1*. These results should be interpreted with caution as no correction for multiple comparisons was performed, and similar analyses should be performed on a larger cohort of PD patients. However, we propose a model in which the risk for PD may be driven not only by the number of repeats, but also by the genotypes of SNVs within the risk-haplotype, affecting *C9orf72* RNA expression levels. Further studies are needed to elucidate the possible role of *C9orf72* in PD pathogenesis.

## Figures and Tables

**Figure 1 genes-12-01210-f001:**
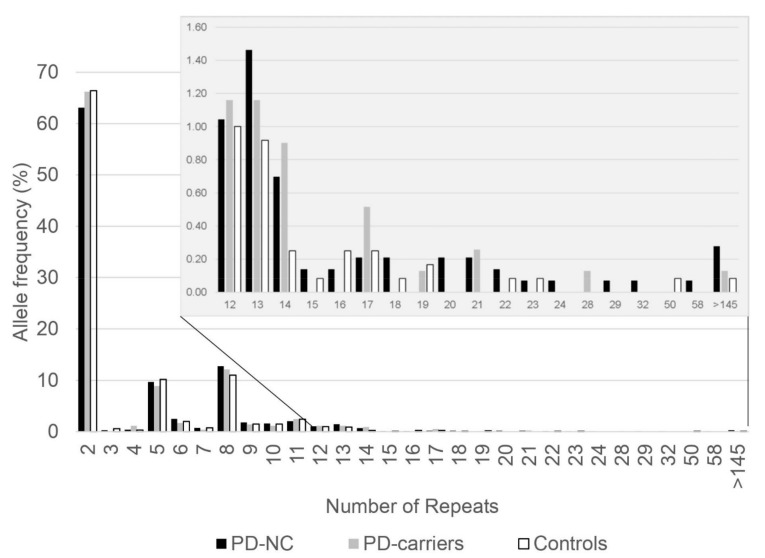
Graphic representation of *C9orf72* G_4_C_2_ hexanucleotide repeat allele frequencies in Ashkenazi Parkinson’s disease patients and controls. The repeats’ allele frequencies for each group are presented: PD patients without *LRRK2*, *GBA*, and *SMPD1* mutations (PD-non-carriers, black); PD patients carrying mutation in *LRRK2*, *GBA*, or *SMPD1* (PD-carriers, gray); and controls as previously published (Reference [26], white). Upper panel is a zoomed-in graph of 12 repeats and higher.

**Figure 2 genes-12-01210-f002:**
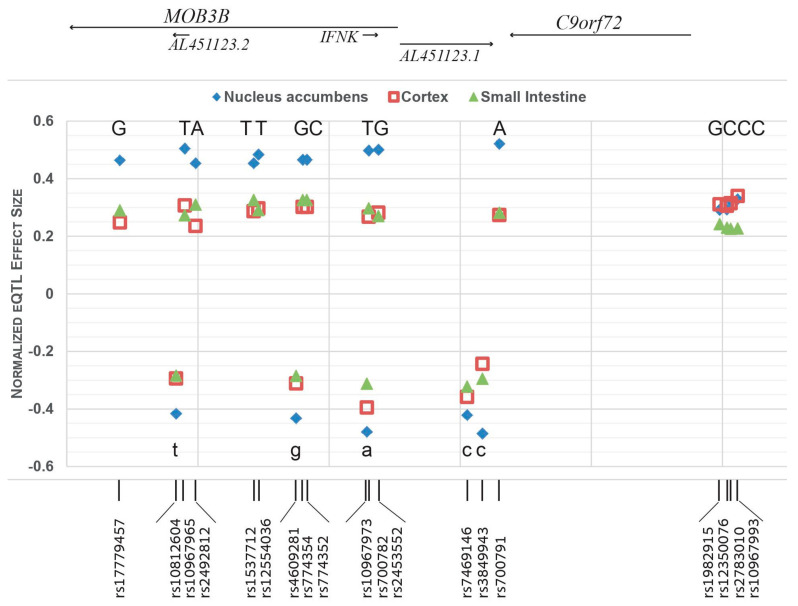
Higher levels of *C9orf72* RNA expression for all 19 SNVs within the risk-haplotype: Normalized eQTL effect size in Nucleus accumbens (dimond), Cortex (square), and small intestine (triangle). Upper panel and upper-case letters are the alternate allele observed in the risk haplotype, associated with higher levels of *C9orf72* expression; lower panel and lower-case letters are the reference allele observed in the risk haplotype, associated with higher levels of *C9orf72* expression.

**Figure 3 genes-12-01210-f003:**
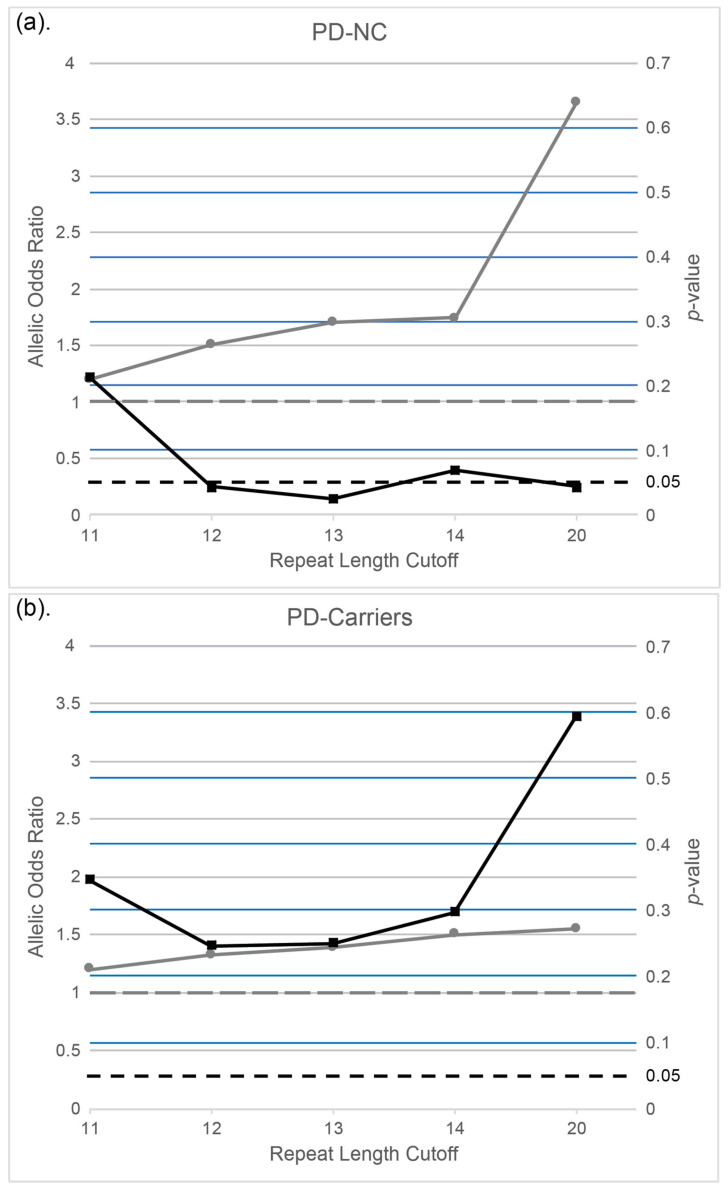
The effect of each cutoff of *C9orf72*-repeats-size on Parkinsons’ disease risk in non-carriers of mutations ((**a**), PD-NC) and in carriers of mutation ((**b**), PD-Carriers; see Methods). Grey lines (circles) depict the Odds ratio and black lines (squares) depict the *p*-value. Dashed grey lines represent Odd Ratio of 1.0, and dashed black lines represent *p*-value of 0.05.

**Table 1 genes-12-01210-t001:** Characteristics of the 1106 Parkinson’s disease patients of Ashkenazi Jewish origin.

	Non-Carrier PD Patients ^a^	PD Patients Carriers of *LRRK2*, *GBA*, or *SMPD1* Mutations
N	718	388 ^c^
Women, N (%)	266 (37.0)	171 (44.1)
AAE, mean (SD), y	68.5 (10.2) ^d^	65.4 (10.1)
AAO, mean (SD), y	61.4 (11.5)	58.4 (10.6)
Family history of PD ^b^, N (%)	144 (20)	120 (30.9)

Abbreviations: PD, Parkinson disease; N, number of individuals; AAE, age at enrollment; SD, standard deviation; y, years; AAO, age at disease onset. ^a^ Patients without the *LRRK2*, *GBA*, or *SMPD1* mutations (specified in the Methods section). ^b^ Patients with 1st or 2nd degree family members with PD. ^c^ Carrier patients included: 133 individuals with *LRRK2* mutation, 223 individuals with *GBA* mutations, 8 individuals with *SMPD1* mutation, 23 individuals with mutations in both *LRRK2* and *GBA*, and 1 patient with both *GBA* and *SMPD1* mutations. ^d^ Data regarding AAE was not available for one individual.

**Table 2 genes-12-01210-t002:** The association 2–60 *C9orf72* hexanucleotide repeats with Parkinson’s disease.

Cohort	Non-Carrier PD Patients ^a^	PD Patients Carriers of *LRRK2*, *GBA*, or *SMPD1* Mutations	Controls ^b^
2–19 repeats, N (%)	701 (98.2)	384 (99.2)	596 (99.5)
20–60 repeats, N (%)	13 (1.8)	3 (0.8)	3 (0.5)
Odds Ratio (95% CI)	3.684 (1.045–12.990)	1.552 (0.312–7.729)	
*p*-value ^c^	0.041	0.684	

Abbreviations: PD, Parkinson’s disease; N, Number of individuals; CI, confidence interval. The longest allele in each individual was recorded. ^a^ Patients without the *LRRK2*, *GBA*, or *SMPD1* mutations (specified in the Methods section). ^b^ The values in the control cohort were previously published [26]. ^c^ 2 × 2 Fisher’s exact test (2-sided).

**Table 3 genes-12-01210-t003:** The effect of the 19 variations within the risk-haplotype on RNA expression levels and splice variants as reported by GTEx database.

Location, chr9 (hg38)	rs ID	Gene	Risk Allele	Ref > Alt ^a^	Highest eQTL for *C9orf72*, NES (Tissue)	eQTL for *C9orf72* in Cerebellum, NES	Highest sQTL for *C9orf72*, NES (Tissue)	GnomAD v3.1- AF for the Risk Allele (in Non-Neuro Cases)		
								AJ	European (Non-Finnish)	All Populations
**27488094**	**rs17779457**	***MOB3B***	**G**	**T > G**	**0.464 (N.A.)**	**0.341**	−0.88 (Cer)	0.2097	0.2428	0.2490
27496663	rs10812604	*MOB3B*	**T**	**T > G**	−0.416 (N.A.)	−0.343	0.70 (Cer)	0.2252	0.2861	0.3555
27497990	rs10967965	*MOB3B*	**T**	**A > T**	0.505 (N.A.)	0.360	−0.96 (C.H.)	0.1742	0.2185	0.1547
27499629	rs2492812	*MOB3B*	**A**	**C > A**	0.454 (N.A.)	0.334	−0.88 (C.H.)	0.2091	0.2419	0.2486
27508491	rs1537712	*MOB3B*	**T**	**C > T**	0.454 (N.A.)	0.332	−0.75 (Cer)	0.2323	0.2753	0.2915
27509213	rs12554036	*MOB3B*	**T**	**G > T**	0.484 (N.A.)	0.345	−1.0 (C.H.)	0.2054	0.2335	0.1796
27514964	rs4609281	*MOB3B*	**G**	**G > T**	−0.432 (N.A.)	−0.364	0.71 (Cer)	0.2417	0.2865	0.3509
27515969	rs774354	*MOB3B*	**G**	**A > G**	0.466 (N.A.)	0.345	−0.80 (C.H.)	0.2354	0.2754	0.2909
27516592	rs774352	*MOB3B*	**C**	**T > C**	0.466 (N.A.)	0.345	−0.80 (C.H.)	0.2354	0.2755	0.2910
27525753	rs10967973	*IFNK*	**A**	**A > G**	−0.602 (Cer)	−0.602	0.66 (Cer)	0.3570	0.4640	0.5599
27526049	rs700782	*IFNK*	**T**	**C > T**	0.498 (N.A.)	0.370	−0.98 (C.H.)	0.2109	0.2440	0.2461
27527514	rs2453552	*MOB3B*	**G**	**T > G**	0.501 (N.A.)	0.36	−1.0 (C.H.)	0.2110	0.2450	0.2650
27541043	rs7469146	*C9orf72*	**C**	**C > T**	−0.542 (Cer)	−0.542	0.65 (Cer)	0.3736	0.5119	0.5916
27543384	rs3849943	*C9orf72*	**C**	**C > T**	−0.485 (N.A.)	−0.336	0.96 (Cer, C.H.)	0.2005	0.2362	0.2178
27545962	rs700791	*C9orf72*	**A**	**C > A**	0.521 (N.A.)	0.388	−1.1 (C.H.)	0.1958	0.2293	0.1979
27579562	rs1982915	Intergenic	**G**	**A > G**	0.407 (Cer)	0.407	−0.49 (Cer)	0.4212	0.4982	0.4745
27580676	rs12350076	Intergenic	**C**	**A > C**	0.408 (Cer)	0.408	−0.51 (Cer)	0.4262	0.5104	0.4964
27581241	rs2783010	Intergenic	**C**	**T > C**	0.418 (Cer)	0.418	−0.56 (Cer)	0.4273	0.4645	0.4808
27582313	rs10967993	Intergenic	**C**	**T > C**	0.424 (Cer)	0.424	−0.54 (F.C.)	0.4025	0.5044	0.5032

^a^ In bold is the allele that is associated with higher *C9orf72* RNA expression levels, as reported by GTEx database. Ref = reference allele; Alt = Alternate allele; eQTL = expression quantitative trait loci; sQTL = spliced quantitative trait loci; NES = normalized effect size; AF = Allele frequency; AJ = Ashkenazi Jews; N.A. = Nucleus accumbens; Cer = Cerebellum; C.H. = Cerebellar hemisphere; F.C. = Frontal Cortex.

## Data Availability

The data presented in this study are available in the Appendix A.

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
