# Peer review of "C9orf72-G4C2 Intermediate Repeats and Parkinson’s Disease; A Data-Driven Hypothesis"

_genes, 2021, doi:10.3390/genes12081210_

Round 1

Reviewer 1 Report

Previous efforts have hinted possible association between the length of non-pathogenic repeat size on C9orf72 with PD risk.  In this report, authors provide the analysis of Ashikenazi Jewish (AJ) cohort showing that possible increased PD risk with increase in C9orf72 repeat length.  Thus, current report is an additional report suggesting that intermediate C9orf72 repeats may increase risk for parkinson's disease.  While the authors do show that these intermediate alleles lead to increased expression of C9orf72, the authors really do not provide additional insights regarding the validity of C9orf72 with PD risk.  I also following specific recommendations that might increase the utility of this report.

  1. GWAS has identified other significant risk factors for PD, including SNCA polymorphism and MAPT haplotype. What is the risk associated with these other factors in the current cohort.
  2. The overall odds ratio seem rather high compared with other comparable studies.  This should be discussed further.
  3. It is unclear if the cohorts were also screened for possible mutations in recessive genes, such as PRKN.

Author Response

We thank the reviewer for the very important and insightful comments, and we hereby describe a detailed-point-by-point reply (in bold) to all comments and suggestions.  

Response to reviewer 1

  1. GWAS has identified other significant risk factors for PD, including SNCA polymorphism and MAPT haplotype. What is the risk associated with these other factors in the current cohort.

We thank the reviewer for this important comment. Of interest, our previous GWAS analysis of Ashkenazi (AJ) Parkinson’s disease patients (Vacic et al. HMG 2014, reference no. 29 in the manuscript) demonstrated a hit at the MAPT locus. However, this hit suggested a protective effect (OR=0.62). In addition, the hit at the SNCA locus did not reach significance at the multiple correction level. Therefore, our AJ patient cohort was not screened for founder mutations in these genes, though we cannot entirely exclude the possibility that a rare mutation could exist.

  1. The overall odds ratio seem rather high compared with other comparable studies.  This should be discussed further.

We thank the reviewer for this comment. We believe that the relatively genetically homogenous population we are studying, the Ashkenazi Jews (AJ), can identify PD risk alleles otherwise missed in heterogeneous populations. The religious and cultural constrains of the AJ increased the frequencies of several rare alleles. This phenomenon was already well documented for the LRRK2-G2019S and GBA-N370S alleles in AJ-Parkinson’s disease patients. Moreover, we analyzed the results in a stratified manner, separating the LRRK2/GBA/SMPD1 carriers from the non-carriers, which increased the odds ratio to 3.68 for the intermediate repeats of 20-26.

We therefor added to the Discussion section in the revised manuscript a new sentence, in page 9, lines 258-261, as followed:

“The high significant odds ratio of 3.68 in non-carriers may be due to the exclusion of those patients who carry known risk alleles in LRRK2, GBA, or SMPD1, as we believe that in these patients the risk for PD is likely influenced by these mutations and not by the C9orf72 intron 1 hexanucleotide repeat numbers.”

  1. It is unclear if the cohorts were also screened for possible mutations in recessive genes, such as PRKN.

As in our answer to question 1, our GWAS and whole-genome-sequencing studies on a portion of the cohort participating in the C9orf72 study showed no significant hits for PRKN locus. With that said, we cannot exclude the possibility that some of our “non-carriers” might carry rare alleles in this gene, but these cases should be extremely rare.

Reviewer 2 Report

Genes-1309410

The aim of this study was to assess the impact of repeat expansions in C9orf72, typically associated with ALS, on the risk of developing PD in the Ashkenazi Jewish population. The authors report a significant association between C9orf72 repeats (intermediate length) and PD, but only in the group not carrying the specific mutations in other genes. The authors then went on to define a 19 allele risk haplotype in the C9orf72 locus, with all 19 alleles being associated with higher expression. Overall this is a very interesting manuscript describing a genetic association in PD with a gene not normally linked to PD, but for which evidence is mounting that it can play a role in other neurodegenerative diseases. There are a few issues to be addressed prior to publication, outlined below:

Specific points:

  1. The authors could explain in 1 or 2 additional sentences the rational for stratifying the PD cases in this manner (carriers of different mutations with PD vs PD patients without the specific mutations).
  2. On page 5, in the risk-haplotype in C9orf72, are each of the 44 SNVs individually associated with expanded intron 1 repeats, or is this a collective association?
  3. Were these genotype analyses (the risk-haplotype analyses) stratified to carrier vs non-carrier as in the previous sections, as was performed in assessing the PD risk?
  4. On page 7, a brief sentence describing the rationale for selecting 14 repeats as the best assessor for PD risk haplotype.
  5. What could be the potential mechanism(s) for the divergence between ALS and PD pathogenesis in the context of C9orf72 repeat expansion? If the risk, at least for PD, is also influenced by overall expression levels, then does this indicate that motor neurons may be more sensitive to reduced expression of C9orf72?
  6. The transcriptional changes in cells with low vs intermediate repeat lengths involve systems and pathways that are not only implicated in PD, but also ALS, can the authors speculate as to other potential mechanisms?

Author Response

We thank the reviewer for the very important and insightful comments, and we hereby describe a detailed-point-by-point reply (in bold) to all comments and suggestions. 

Response to reviewer 2

  1. The authors could explain in 1 or 2 additional sentences the rational for stratifying the PD cases in this manner (carriers of different mutations with PD vs PD patients without the specific mutations).

We thank the reviewer for this comment. As the known Founder Ashkenazi Jewish (AJ) risk alleles in the GBA gene (N370S, 84GG and others) and the known LRRK2-G2019S risk allele are observed in the AJ in higher frequencies than in other populations (and in about 34% of our AJ-PD patients; see reference 28 in the manuscript), we wanted to evaluate the effect of C2orf72 intermediate repeats on a group of PD patients who do not carry known risk alleles. Among the 1106 AJ-PD patients studied in this report, 388 carried mutations in either LRRK2, GBA, or SMPD1 (35.1%). We did not want the number of C9orf72 repeats in this large group of patients to mask the effect in non-carriers. We added this explanation in the results section, subsection 3.1., page 3, lines 119-122 in the revised manuscript:

“We ran a stratified analysis based on the carrier status in LRRK2, GBA, and SMPD1, as a high percentage of our PD cohort carry risk alleles in these 3 genes (35.1%, 388/1106), and these carrier-patients may mask the effect of the hexanucleotide repeat length on PD-risk in non-carrier patients.”

2. On page 5, in the risk-haplotype in C9orf72, are each of the 44 SNVs individually associated with expanded intron 1 repeats, or is this a collective association?

This is a collective association. The association is for the complete haplotype.

3. Were these genotype analyses (the risk-haplotype analyses) stratified to carrier vs non-carrier as in the previous sections, as was performed in assessing the PD risk?

Yes. The risk haplotype analysis was also done in a stratified manner. In the Results section, subsection number 3.3., bottom of page 6, lines 199-205 in the revised manuscript, we wrote: “Overall enrichment of the risk-haplotype was observed in PDs compared to controls: 167 out of 588 PDs carried one or two copies of the risk-haplotype (28.4%) compared to 24 out of 126 controls (19.0%). When stratifying based on mutation carrier status (PD-carriers and PD-NC), a significant association was detected in PD-NC: 28.6% of them carried one or two copies of the risk-haplotype (OR=1.71, CI=1.04-2.81, p=0.0356), and tendency was shown in PD-carriers (28.0%, OR=1.65, CI=0.97-2.82, p=0.0656).”

4. On page 7, a brief sentence describing the rationale for selecting 14 repeats as the best assessor for PD risk haplotype.

We thank the reviewer for this comment. We have clarified the language to make the rationale clearer.

We changed the sentence in page 7:

We deleted lines 210-212 in the original manuscript, which stated:

“Using 14 repeats as the best assessor for carrying the risk-haplotype, we re-calculated the association of 14 repeats and higher with PD in our cohorts.”

Instead, we inserted the following sentence in lines 212-214 of the revised manuscript, which states now:  

Thus, we used 14 repeats as the best assessor for carrying the risk-haplotype, and re-calculated the association of 14 repeats and higher with PD in our cohorts.”

5. What could be the potential mechanism(s) for the divergence between ALS and PD pathogenesis in the context of C9orf72 repeat expansion? If the risk, at least for PD, is also influenced by overall expression levels, then does this indicate that motor neurons may be more sensitive to reduced expression of C9orf72?

  • 5.1 What could be the potential mechanism(s) for the divergence between ALS and PD pathogenesis in the context of C9orf72 repeat expansion?

We thank the reviewer for this very important comment. This is indeed an intriguing biological question, yet to be answered for PD patients. Our hypothesis is that in ALS, when the number of repeats is very high (more than 100 repeats) the expansion results in reduced gene expression or even a null effect on the gene expression, via a methylation pathway, that can translate to haploinsufficient levels of C9orf72. More importantly, RNA foci and dipeptide-repeats (DPR) aggregations synthesized from sense and antisense expanded repeat-containing transcripts add a negative dominant effect on the cells. However, with intermediate repeats, as shown in Cali et al. (reference 38 in our manuscript), these negative effects are not seen (no RNA foci or DPR were observed in intermediate repeats), and in contrast to haploinsufficiency with large expansions, an increase of gene expression is observed.

  • 5.2 If the risk, at least for PD, is also influenced by overall expression levels, then does this indicate that motor neurons may be more sensitive to reduced expression of C9orf72?

The reviewer raises an interesting and important assumption regarding the possible mechanism involved. At this stage, there are no data that can answer this specific question about cell specificity. We believe that further studies that will employ single cell analyses on both motor neuron-induced PS cells (iPSc) from PD patients and on tissue models will likely help verify this possibility.  

6. The transcriptional changes in cells with low vs intermediate repeat lengths involve systems and pathways that are not only implicated in PD, but also ALS, can the authors speculate as to other potential mechanisms? 

Indeed, in the Discussion section of the manuscript, we suggest several potential pathways that could be involved, such as vesicle trafficking and protein degradation pathways, including Golgi vesicle transport, response to ER-stress and autophagy pathways, based also on Cali et al. (reference 38). This is detailed in page 10, lines 303-308 in the revised manuscript. It is also possible that additional pathways are involved, as Cali et al. detected 59 shared dysregulated genes when comparing between the two states of intermediate repeats and expansion repeats states, relative to a low number of repeats’ state. Of them, 24 changed expression direction (17 from up to down, and 7 from down to up). However, the identities of those genes were not disclosed in the publication of Cali et al. (reference 38). It is clear that further studies are necessary to shed light on these cellular mechanisms involved in the overall effect of C9orf72 hexanucleotide repeats length.